# Vitamin D and Diseases of Mineral Homeostasis: A *Cyp24a1* R396W Humanized Preclinical Model of Infantile Hypercalcemia Type 1

**DOI:** 10.3390/nu14153221

**Published:** 2022-08-06

**Authors:** René St-Arnaud, Alice Arabian, Dila Kavame, Martin Kaufmann, Glenville Jones

**Affiliations:** 1Research Centre, Shriners Hospital for Children-Canada, Montreal, QC H4A 0A9, Canada; 2Department of Human Genetics, Faculty of Medicine and Health Sciences, McGill University, Montreal, QC H3A 0C7, Canada; 3Department of Surgery, Faculty of Medicine and Health Sciences, McGill University, Montreal, QC H3G 1A4, Canada; 4Department of Medicine, Faculty of Medicine and Health Sciences, McGill University, Montreal, QC H3A 1A1, Canada; 5Department of Biomedical and Molecular Sciences, Queen’s University, Kingston, ON K7L 3N6, Canada; 6Department of Surgery, Queen’s University, Kingston, ON K7L 3N6, Canada

**Keywords:** infantile hypercalcemia type 1, idiopathic infantile hypercalcemia, CYP24A1, 25-hydroxyvitamin D 24-hydroxylase, hypercalcemia, 24,25-dihydroxyvitamin D_3_, fracture repair

## Abstract

Infantile hypercalcemia type 1 (HCINF1), previously known as idiopathic infantile hypercalcemia, is caused by mutations in the 25-hydroxyvitamin D 24-hydroxylase gene, *CYP24A1*. The R396W loss-of-function mutation in *CYP24A1* is the second most frequent mutated allele observed in affected HCINF1 patients. We have introduced the site-specific R396W mutation within the murine *Cyp24a1* gene in knock-in mice to generate a humanized model of HCINF1. On the C57Bl6 inbred background, homozygous mutant mice exhibited high perinatal lethality with 17% survival past weaning. This was corrected by crossbreeding to the CD1 outbred background. Mutant animals had hypercalcemia in the first week of life, developed nephrolithiasis, and had a very high 25(OH)D_3_ to 24,25(OH)_2_D_3_ ratio which is a diagnostic hallmark of the HCINF1 condition. Expression of the mutant *Cyp24a1* allele was highly elevated while *Cyp27b1* expression was abrogated. Impaired bone fracture healing was detected in CD1-R396^w/w^ mutant animals. The augmented lethality of the C57Bl6-R396W strain suggests an influence of distinct genetic backgrounds. Our data point to the utility of unique knock-in mice to probe the physiological ramifications of CYP24A1 variants in isolation from other biological and environmental factors.

## 1. Introduction

Idiopathic infantile hypercalcemia (IIH) was first reported by Lightwood in the 1950s in the United Kingdom. An elevation in the number of IIH cases was observed following the introduction of an increased prophylactic dose of vitamin D in infant formula and milk products [1,2]. The infants were found to have high levels of serum and urinary excretion of calcium, physiological levels of serum phosphate and magnesium, and occasionally low serum alkaline phosphatase (ALP). Reducing the vitamin D dose by half lowered the observed incidence. Because some infants were not affected by treatment, Lightwood later hypothesized that the disease was caused by fluctuation in infants’ sensitivity to vitamin D.

The general manifestations of this rare disease are high serum calcium and high urinary calcium loss, failure to thrive, vomiting, constipation, polyuria with dehydration, and Ca deposits in the kidney, with susceptibility to renal complications such as nephrolithiasis or nephrocalcinosis. Affected infants may also manifest muscular hypotonia and lethargy, which leads to growth impairment, with delays in mental and mobility development [3]. Affected adults can also manifest some extra-renal pathologies, such as Ca deposits in the joints or the cornea, and low bone mineral density with osteoporosis, mainly due to enhanced osteoclastic activity [4,5]. The genetic cause of IIH remained unknown until Schlingmann and coworkers identified it in 2011. They found inactivating mutations in *CYP24A1* that drive the disease evolution [6].

Since mutations in the *SLC34A1* (solute carrier family 34 member 1, a type II sodium-phosphate cotransporter) gene cause a similar clinical presentation [7], the nomenclature has been changed to infantile hypercalcemia type 1 for *CYP24A1* mutations (HCINF1, OMIM 143880) and infantile hypercalcemia type 2 for *SLC34A1* affected patients (HCINF2, OMIM 616963).

To date, more than 50 disease-causing mutations of *CYP24A1* have been described in the literature [8,9,10,11,12,13]. Alterations of the *CYP24A1* sequence can impact CYP24A1 structure and activity in different ways. Mutations are predicted to alter the substrate or heme binding site, the binding of adrenodoxin, access of the substrate or exit of the product, or generally to affect protein folding [6,14,15,16,17].

The currently available *Cyp24a1* knockout strain [18] developed more than 20 years ago exhibits hypercalcemia, hypercalciuria, and nephrocalcinosis and could represent a model of HCINF1. Supporting this view, it was shown that surviving *Cyp24a1*-null animals possess a much-reduced ability to clear a bolus dose of [1β-^3^H]1α,25-(OH)_2_D_3_ compared with wild-type littermates [19]. However, it should be noted that the *Cyp24a1*-deficient strain was generated with a selection cassette inserted in the opposite orientation to replace the heme-binding exon [18]. The *Cyp24a1*-deficient mice from this strain does not express any *Cyp24a1* message [18]. This contrasts with the situation in patients who, in many cases, express the mutated *CYP24A1* mRNA and protein [16]. Moreover, some of the *CYP24A1* mutations identified in HCINF1 patients are hypomorphic mutations, demonstrating residual or altered activity [6]. There is thus a need to generate improved preclinical animal models of infantile hypercalcemia type 1 based on human mutations to better understand the effects of dysfunctional vitamin D metabolism in HCINF1 patients.

We have introduced the site-specific R396W mutation within the murine *Cyp24a1* gene in knock-in mice. The R396W loss-of-function mutation is the second most frequent mutated allele observed in affected HCINF1 patients [8]. Profiling vitamin D metabolites using a sensitive LC-MS/MS-based assay demonstrated that the R396W strain is a valid preclinical model of HCINF1.

## 2. Materials and Methods

### 2.1. Generation of the R396W Knock-in Strain

All animal procedures were reviewed and approved under Animal Use Protocol number 7470 by the Shriners Hospitals for Children—Canada Institutional Animal Care and Use Committee and followed the guidelines of the Canadian Council on Animal Care. Mice were maintained in an environmentally controlled barrier animal facility with a 12 h light, 12 h dark cycle, and had access to mouse chow and water ad libitum.

The R396W knock-in embryonic stem cells were generated by Cyagen Biosciences (Santa Clara, CA, USA) under contract. A confirmed targeted clone was injected into C57Bl6/N blastocysts. Chimeras produced by blastocyst injection of targeted embryonic stem cells transmitted the knock-in mutation to their progeny and heterozygous mice were interbred to obtain all three genotypes (wild-type: C57Bl6-R396^+/+^; heterozygous: C57Bl6-R396^+^^/w^; homozygous mutant: C57Bl6-R396^w/w^). For phenotyping, mice were sacrificed at postnatal Day 7 (P7) and at 4 months of age.

The C57Bl6-R396W strain was also outbred to CD1 mice to generate the CD1-R396W strain. The phenotype of these mice was analyzed at 3 months of age.

### 2.2. Genotyping

All mice were genotyped using genomic DNA isolated from carcasses found in cages prior to weaning or from ear punches at 3 weeks of age. For the R396W strains, the different genotypes were determined using a custom TaqMan single-nucleotide polymorphism (SNP) genotyping assay from Applied Biosystems (Life Technologies, Foster City, CA, USA) with *Cyp24a1* primers (forward: 5′-GCTTACCCCAAGTGTGCCATT-3′ and reverse: 5′-CCAGAACGGTTGGCTTGTC-3′) flanking the single-point mutation site. The 396R SNP primer (VIC-conjugated) was 5′-AAGGGTCCGAGTTGTG-3′ and the 396W SNP primer (FAM-conjugated) was 5′-AAGGGTCCAAGTTGTG-3′ (mutated nucleotide underlined).

### 2.3. Survival Analysis

Cages were monitored daily until weaning (day 21) and when found, cadavers/remains were collected for genotyping. The percent survival curves per genotype were graphed using GraphPad Prism (San Diego, CA, USA) version 7.04.

### 2.4. Serum Biochemistry

Total serum calcium was measured using an automated analyzer. Serum levels of vitamin D metabolites were assayed by LC-MS/MS following 4-(2-(6,7-dimethoxy-4-methyl-3,4-dihydroquinoxalinyl)ethyl)-1,2,4-triazoline-3,5-dione (DMEQ-TAD) derivatization as described previously [20]. Briefly, serum aliquots and calibrators were diluted 1:3 with water and spiked with internal standards. Proteins were precipitated by sequentially adding 0.1 M HCl, 0.2 M zinc sulfate, and methanol, with vortexing after the addition of each component. Tubes were centrifuged 10 min at 12,000× *g* and supernatants were transferred to borosilicate glass tubes. Organic extraction was carried out by adding equal volumes of hexane and methyl tertiary butyl ether with vortexing after the addition of each component. The upper organic phase was transferred into LC-MS/MS sample vials and evaporated under nitrogen flow. Dried residues were derivatized by addition of 0.1 mg/mL DMEQ-TAD dissolved in ethyl acetate for 30 min at room temperature in the dark, then a second time for 60 min. The reaction was stopped by addition of ethanol, samples were dried and redissolved in 60:40 methanol:water running buffer. LC-MS/MS analysis was performed using an Acquity UPLC connected in line with a Xevo TQ-S mass spectrometer in electrospray positive mode (Waters). Chromatographic separations were achieved using a BEH-Phenyl UPLC column (1.7 µm, 2.1 × 50 mm) (Waters) and methanol/water-based gradient solvent system. Simultaneous assay of 1,25(OH)_2_D_3_ and 1,24,25(OH)_3_D_3_ was possible based on cross-reactivity of an anti-1,25(OH)_2_D_3_ antibody slurry (Immundiagnostik, Manchester, NH) with 1,24,25(OH)_3_D_3_. The serum was incubated with 100 µL of anti-1,25(OH)_2_D_3_ antibody slurry [21] for 2h at room temperature with orbital shaking at 1200 rpm. The slurry was isolated by vacuum filtration and vitamin D metabolites were eluted, derivatized with DMEQ-TAD, and separated using a longer LC step as previously described [22,23].

### 2.5. Microcomputed Tomography of Whole Kidneys

Kidneys were harvested from mice at P7 and fixed in 4% paraformaldehyde diluted in PBS for 24 h, washed two to three times, and then stored in 70% ethanol. Kidneys were scanned with high-resolution microcomputed tomography using a SkyScan model 1272 scanner (Bruker, Kontich, Belgium).

### 2.6. Gene Expression Monitoring

Kidneys were harvested at P7 and 4 months, dissected free of surrounding tissue, immersed in RNA later (Ambion, Austin, TX, USA), and stored at −80 °C until ready for testing. Quantitative gene expression was assessed by real-time reverse transcriptase polymerase chain reaction (RT-qPCR). Briefly, kidneys were homogenized in 1 mL TRIzol reagent (Invitrogen, Carlsbad, CA, USA) and total RNA was isolated according to the manufacturer’s protocol. One (1) µg of RNA was reverse transcribed to cDNA using High-Capacity cDNA Reverse Transcription Kit (Life Technologies Applied Biosystems, Waltham, MA, USA). Real-time qPCR using TaqMan universal PCR master mix and gene-specific Taqman assays for *Cyp24a1* (Mm00487244_m1), *Cyp27b1* (Mm01165918_m1), vitamin D receptor (VDR, Mm01309608_m1), *Casr* (Mm00443375_m1), Calbindin D9k (*S100g*, Mm00486654_m1), Calbindin D28k (*Calb1*, Mm00486645_m1), *Trpv5* (Mm01166030_m1), *Trpv6* (Mm00499069_m1), *Npt2a* (*Slc34a1*, Mm00441450_m1), *Npt2b* (*Slc34a2*, Mm01215846_m1), *Npt2c* (*Slc34a3*, Mm00551746_m1), *Npt3* (*Slc17a2*, Mm00522866_m1), *Nkcc2* (*Slc12a1*, Mm01275821_m1), and *Nhe3* (*Slc9a3*, Mm01352473_m1) was performed using a QuantStudio 7 real-time PCR system (Life Technologies Applied Biosystems). The assay was performed in triplicate. Relative quantification of mRNA was performed according to the comparative C_t_ method and normalized to housekeeping genes.

### 2.7. Intramedullary Rodded Tibial Osteotomy

The surgical procedure was performed on mice under general isoflurane anesthesia as described in Martineau et al. [24]. Briefly, an incision was performed above the right knee to free the patellar ligament from lateral tissue. A 25G spinal needle wire guide was inserted down the medullary canal through a 26G needle pushed through the tibial plateau. The wire guide was bent at a right angle, cut at the tibial plateau, and secured by a mattress suture on either side of the patellar ligament after the needle was pulled out. The tibial shaft was cut using micro scissors about 2–3 mm above the tibiofibular junction. Topical analgesics were applied at the wound site, then the skin was sutured. Carprofen was provided at the time of surgery and for the following 48 h post-osteotomy. The mice were sacrificed using isoflurane and CO_2_ at day 10 (D10) post-surgery.

### 2.8. Three-Point Bending Assays

For three-point bending assays, callus tibiae were thawed overnight at room temperature and tested for mechanical properties using an Instron model 5943 single-column table frame machine (Instron, Norwood, MA, USA). The fractured bones rested on two fulcra set 6 mm apart. Load-sensing cell was applied to the widest part of the callus. Raw output used for comparison was strength (load at break, in N).

### 2.9. Statistical Analysis

Statistical analyses were performed using GraphPad Prism version 7.04. Statistical tests involved 2-tailed *t*-tests for gene expression monitoring, 1-way ANOVA followed by Dunnett’s or Tukey’s post hoc test for calcemia and three-point bending assays, respectively, or 2-way ANOVA with Sidak’s post hoc test for vitamin D metabolites measurements. The statistical significance threshold was set at a *p*-value of less than 0.05.

## 3. Results

### 3.1. Severe Postnatal Lethality in C57Bl6-R396^w/w^ Mutant Animals

The R396W knock-in targeting construct was electroporated into C57Bl6/N embryonic stem cells. The targeting event was confirmed using both PCR screening and Southern blotting. One (1) correctly targeted clone was injected into C57Bl6/N blastocysts, which allowed to derive germline-transmitting chimeras on the homogeneous C57Bl6/N genetic background. Heterozygous progeny was mated inter se to obtain all three genotypes (wild-type: C57Bl6-R396^+/+^; heterozygous: C57Bl6-R396^+^^/w^; homozygous mutant: C57Bl6-R396^w/w^).

The mutant allele was transmitted at the expected Mendelian frequency, but we detected severe postnatal lethality in the homozygous mutant animals. We observed a significant number of dead pups of the C57Bl6-R396^w/w^ genotype from birth, which culminated in a 17% survival rate of homozygous mutant pups from postnatal day 14 onwards (Figure 1). This is a striking difference from the 50% postnatal lethality measured in the global *Cyp24a1*-deficient strain maintained on a mixed genetic background for more than 20 years [18].

### 3.2. Early Postnatal Hypercalcemia in C57Bl6-R396^w/w^ Mutant Animals

We measured serum calcium levels in wild-type, heterozygous and homozygous mutant pups during the second postnatal week (P7 to P10). C57Bl6-R396^w/w^ mutant animals had significantly elevated circulating calcium levels at P7 compared to wild-type and heterozygous control littermates (Figure 2). This was accompanied by the marked formation of kidney stones (nephrolithiasis) (Figure 3). We did not investigate for signs of extra renal calcification. Circulating serum levels in surviving mutant pups past P7 were within the normal range (Appendix A).

### 3.3. Gene Expression Monitoring

We compared kidney tissue gene expression in C57Bl6-R396^w/w^ mutants and wild-type littermates at P7 and 4 months. The R396W mutant *Cyp24a1* allele was highly overexpressed (30-fold increase) in pups (Figure 4A); this was reduced in surviving adult mutant animals but still remained almost 10-fold higher as compared to wild-type littermates (Figure 4B). As observed in the global *Cyp24a1*-deficient animals [18,19], *Cyp27b1* expression was markedly inhibited at all ages (Figure 4A,B). Vitamin D receptor expression was moderately but significantly overexpressed (Figure 4A,B).

The calcium transporters calbindin D9k and calbindin D28k were significantly overexpressed in mutant pups and this normalized for D28k in adults (Figure 4C,D). There were no differences in the expression levels of the calcium-sensing receptor (*Casr*) or the calcium channels *Trpv5* or *6* between genotypes (Figure 4C,D).

In newborn C57Bl6-R396^w/w^ mutants, we measured the increased expression of the sodium-dependent phosphate transport protein 2b (Npt2b, *Slc34a2*) while related transporters Npt2c (*Slc34a3*) and Npt3 (*Slc17a2*) were expressed at lower levels in mutant animals compared to wild-type littermates (Figure 4E). With the exception of Npt3, which was over-expressed in adult homozygous mutant mice, the expression of co-transporters normalized as animals aged to adulthood (Figure 4F).

### 3.4. CD1-R396W Strain

We crossed the C57Bl6-R396W strain to CD1 outbred mice in an attempt to decrease the extreme perinatal lethality associated with carrying the R396W mutant allele on a homogeneous genetic background. In this mixed genetic background, survival of the CD1-R396^w/w^ mutants was improved to 86% (Figure 5). This allowed us to compare vitamin D metabolites between the strains and examine additional phenotypic manifestations.

### 3.5. Vitamin D Metabolites

We used a sensitive LC-MS/MS-based assay to profile multiple serum vitamin D metabolites [20] in the global *Cyp24a1*-null mice, the C57Bl6-R396W strain, and the CD1-R396W line. The mutant genotypes of all three strains exhibited a five- to six-fold increase in circulating 25(OH)D_3_ compared to control genotypes (Table 1). Surprisingly, levels of 24,25(OH)_2_D_3_ remained detectable but were reduced by 3- to 5.4-fold in mutant mice. These changes led to a very high 25(OH)D_3_ to 24,25(OH)_2_D_3_ ratio in homozygote mutants which is a diagnostic hallmark of the infantile hypercalcemia type 1 condition [20]. Concentrations of 25(OH)D_3_-26,23-lactone were at the lower limit of detection of the assay, in accord with the demonstrated 23-hydroxylation activity of CYP24A1 [25]. Levels of 1,24,25(OH)_3_D_3_ were also significantly reduced in homozygous mutant animals sporting the R396W mutation (Table 1). Despite the much reduced CYP24A1 catabolic enzyme activity in mutant mice, the serum concentration of the hormone 1,25(OH)_2_D_3_ remained normal (Table 1) presumably due to much reduced *Cyp27b1* expression noted in Figure 4A,B. This adaptation is presumably necessary to permit the viability of the surviving animals.

### 3.6. Impaired Bone Fracture Healing in CD1-R396^w/w^ Mice

The improved survival of CD1-R396^w/w^ mutants permitted their skeletal analysis. There were no detectable steady-state skeletal phenotypic manifestations in the mutant animals as shown by normal trabecular bone volume, trabecular separation, thickness, and separation (Appendix A). Since we have observed impaired bone fracture repair in global *Cyp24a1*-deficient mice [24], we challenged the CD1-R396^w/w^ mutant animals to heal surgical osteotomies as a model of bone fracture recovery. The intramedullary rodded immobilized fracture surgery was performed on the left tibia at 3 months of age. The analyses of the fractured bones were performed on post-surgery day 10 (D10). The tibiae with calluses were dissected and the intramedullary nail was carefully removed. The biomechanical properties of the repaired calluses were evaluated using the three-point bending test.

At D10 post-surgery, the analysis of the callus showed a significant decrease in load at break (strength) in heterozygous and homozygous mutant mice compared to wild-type littermates (Figure 6).

## 4. Discussion

The global inactivation of *Cyp24a1* in mice confirmed the physiological importance of CYP24A1 in the catabolism of 1,25(OH)_2_D_3_ [18,19] and elucidated the mechanism of action of 24,25(OH)_2_D_3_ in bone fracture healing [24]. As a preclinical model of infantile hypercalcemia type 1, however, it left to be desired, since the mutation involved replacing the heme-binding exon of *Cyp24a1* by the PGK-neo selection cassette [26] in the opposite transcriptional orientation [18], a genetic change that is significantly different from the mutated alleles identified in patients suffering from the disease (reviewed in [8]). We set out to generate a mouse strain expressing a mutated allele of *Cyp24a1* that corresponds to what has been described in patients and decided on one of the first identified mutations [6] which turns out to be the second most frequent mutated allele observed in affected HCINF1 patients, R396W [8].

We first established the mutation on a homogeneous genetic background, and this resulted in a dramatic rate of perinatal lethality. The elevated calcium concentrations in the first week of life may have contributed to the mortality rate. Despite significant changes in the expression of sodium-phosphate cotransporters, circulating phosphate concentrations were not affected by the mutation (Appendix A), similar to what is observed in patients [27,28]. The differences in the expression of calcium and phosphate transporters observed soon after birth normalized in the surviving population that reached adulthood.

As reported for the global *Cyp24a1*-deficient mice [19], the expression of the vitamin D 1α-hydroxylase gene, *Cyp27b1*, was completely inhibited in young and older C57Bl6-R396^w/w^ mutants, most likely as a protection mechanism to avoid hypervitaminosis. Such an explanation is consistent with the fact that young and older C57Bl6-R396^w/w^ mutants have serum 1,25(OH)_2_D_3_ levels within the normal range. Of note was the very high expression of the mutated *Cyp24a1* allele in homozygous mutants. This contrasts with the measurable but diminished expression of *CYP24A1* in affected patients [16].

The R396W mutation is a loss-of-function mutation [6], yet we measured circulating levels of 24,25(OH)_2_D_3_ in both the inbred and outbred R396W mutated strains. These measurable levels of dihydroxylated metabolite could represent the activity of other cytochrome P450s or be due to the migration of interfering dihydroxyvitamin D metabolites with similar retention time and mass spectral characteristics in LC-MS/MS. We estimate this latter probability as very low. On the other hand, it has been reported that cytochrome P450s distinct from CYP24A1 can synthesize 24,25(OH)_2_D_3_, but it remains unclear if this activity, measured in vitro [29], has relevance in vivo. At any rate, the ability to detect circulating levels of 24,25(OH)_2_D_3_ in CYP24A1-deficient mice or patients allows for calculating the ratio of the metabolite to its precursor 25(OH)D_3_, namely serum 25(OH)D_3_:24,25(OH)_2_D_3_ [20,30,31,32]. Clinically, this ratio is a more reliable indicator of HCINF1 due to *CYP24A1* mutations than serum 24,25(OH)_2_D_3_ alone because it eliminates the possibility that the patient might have a low serum 24,25(OH)_2_D_3_ level due to vitamin D deficiency. The high 25(OH)D_3_:24,25(OH)_2_D_3_ ratio measured in R396W homozygote mutants on both inbred and outbred backgrounds aligns with the reports from over 100 HCINF1 patients [20,32]. We thus posit that the R396W knock-in strains are valid preclinical models of HCINF1.

Another phenotypic manifestation of the R396W mouse mutation that mimics what is observed in patients is the presence of kidney stones. Indeed, hypercalciuria and the risk of renal stones continue in most subjects with *CYP24A1* mutations throughout adult life [28,33,34]. In patients, the disorder may in fact first manifest as hypercalciuria and painful kidney stones in adulthood, without childhood symptoms [16,35,36,37,38]. Based on gene frequency studies, Nesterova and colleagues estimate that the frequency of kidney stones in the general population due to *CYP24A1* mutations might be as high as 4–20% [16].

The perinatal lethality phenotype of *Cyp24a1* homozygous mutants exhibits significant variable penetrance depending on the genetic background upon which the mutation is established. Crossbreeding the R396W mutation on an outbred CD1 background most adequately represents the human condition with no obvious complications unless challenged. Exploiting the differential penetrance of the perinatal lethality between the 3 strains of *Cyp24a1* deficiency available could allow for the identification of modifier gene(s) [39] impacting vitamin D and mineral ion homeostasis.

The *Cyp24a1*-deficient mouse strain [18] was an invaluable tool to study the physiological role of 24,25(OH)_2_D_3_ in mammalian fracture repair. *Cyp24a1*^−/−^ mice show suboptimal endochondral ossification during fracture repair with smaller calluses that exhibit inferior biomechanical properties [24]. The strength of the repairing callus was reduced in heterozygous and homozygous CD1-R396^w/w^ mutant mice compared to wild-type littermates. This change was the only phenotypic manifestation that we observed in heterozygous carriers of the mutation. The impaired bone fracture healing observed in CD1-R396^w/w^ mutant animals establishes that two distinct preclinical models of *Cyp24a1* deficiency exhibit bone fracture repair defects. These results support the need to study bone fracture healing in patients with HCINF1.

## 5. Conclusions

Introducing the site-specific R396W mutation within the murine *Cyp24a1* gene in knock-in mice generated a valid preclinical model of infantile hypercalcemia type 1 with the characteristic high 25(OH)D_3_:24,25(OH)_2_D_3_ ratio that is a diagnostic hallmark of the condition.

## Figures and Tables

**Figure 1 nutrients-14-03221-f001:**
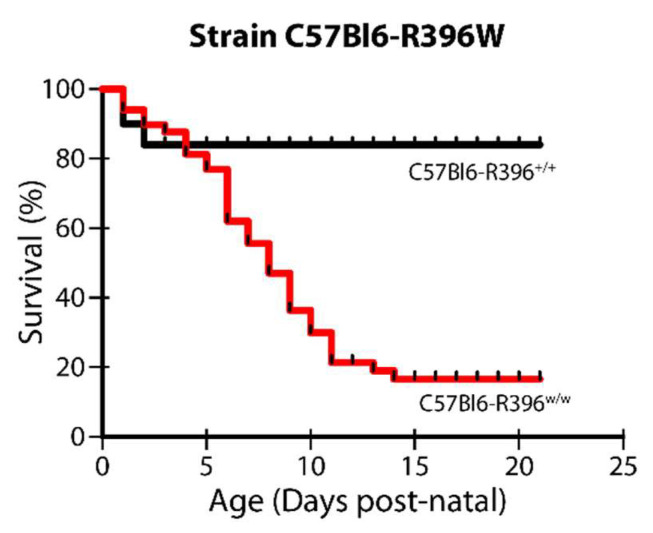
Postnatal lethality in C57Bl6-R396^w/w^ mutant pups. Cages were monitored daily until weaning (day 21) and cadavers were genotyped. Percent survival for wild-type and homozygous mutants is shown.

**Figure 2 nutrients-14-03221-f002:**
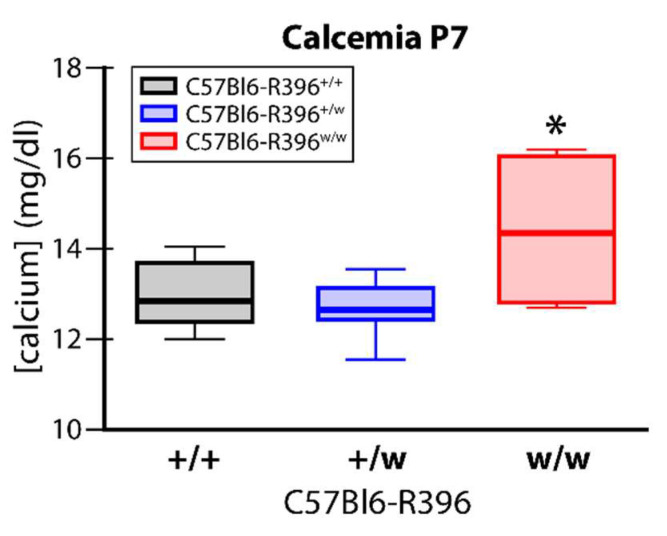
Serum calcium levels in C57Bl6-R396 littermates at postnatal day 7. Total serum calcium was measured using an automated analyzer. +/+, C57Bl6-R396^+/+^; *+*/w, C57Bl6-R396^+^^/w^; w/w, C57Bl6-R396^w/w^. *, *p* < 0.05 by one-way ANOVA and Dunnett’s post hoc test.

**Figure 3 nutrients-14-03221-f003:**
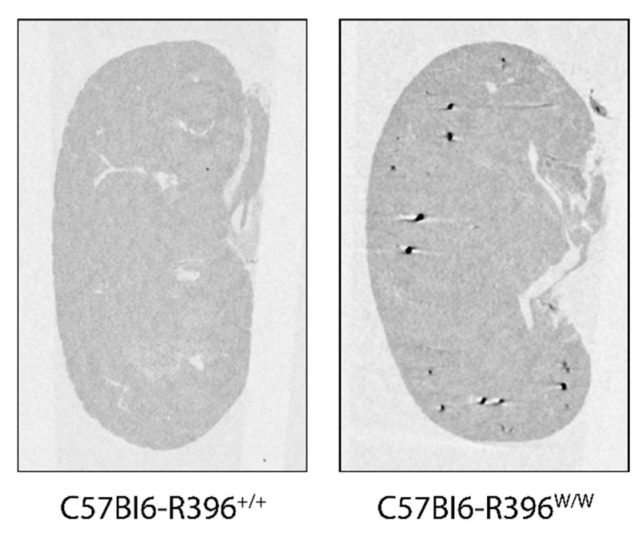
Nephrolithiasis in C57Bl6-R396^w/w^ mutant pups. Kidneys were harvested at P7 and imaged by microCT.

**Figure 4 nutrients-14-03221-f004:**
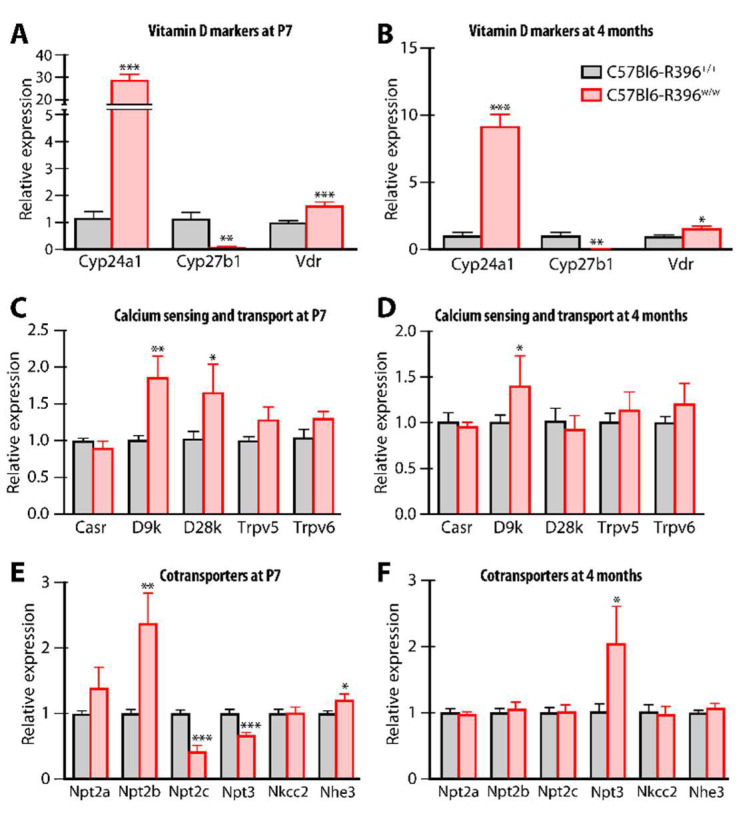
Gene expression monitoring in newborn and adult C57Bl6-R396 littermates. Quantitative gene expression was assessed by RT-qPCR with TaqMan probes on mRNA extracted from kidneys. *, *p* < 0.05; **, *p* < 0.01; ***, *p* < 0.001, double-sided *t*-tests for each gene. (**A**) Vitamin D markers at P7. (**B**) Vitamin D markers at 4 months. (**C**) Calcium sensing and transport at P7. (**D**) Calcium sensing and transport at 4 months. (**E**) Cotransporters at P7. (**F**) Cotransportersat 4 months.

**Figure 5 nutrients-14-03221-f005:**
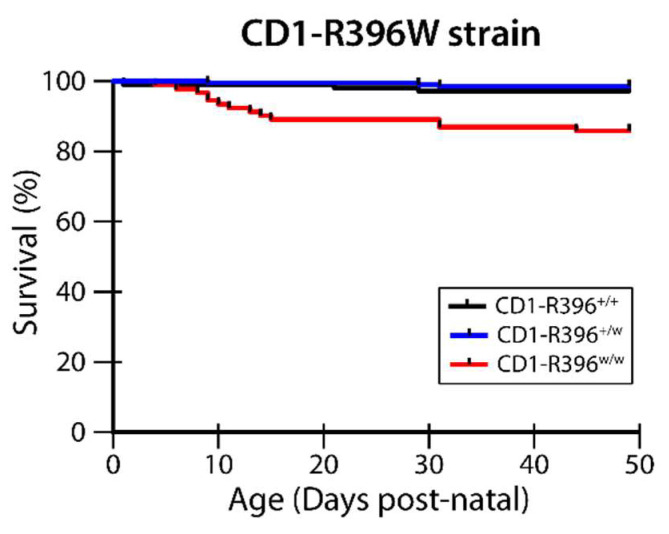
Survival of CD1-R396^w/w^ mutant pups. Cages were monitored daily until postnatal day 49 and cadavers found in cages were genotyped. Percent survival for all 3 genotypes is shown.

**Figure 6 nutrients-14-03221-f006:**
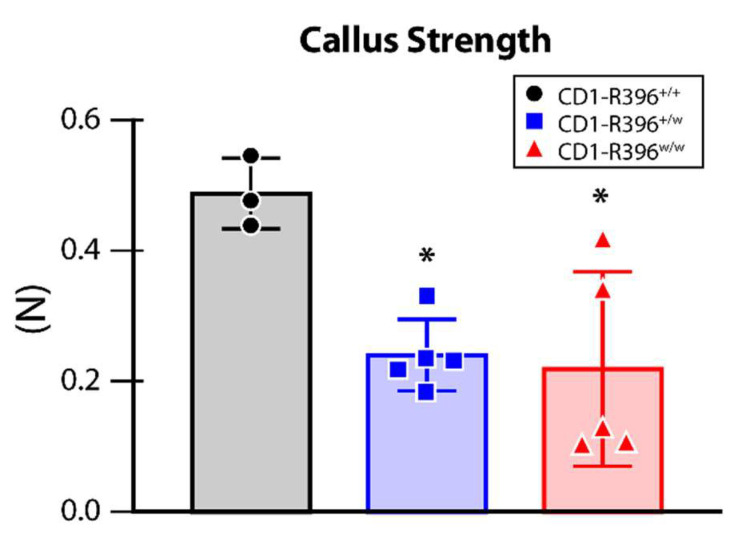
Impaired bone fracture healing in CD1-R396^w/w^ mutant animals. Callus strength (load at break) was calculated from the three-point bending test. *, *p* < 0.05 by one-way ANOVA with Tukey’s post hoc test.

**Table 1 nutrients-14-03221-t001:** Vitamin D metabolites in preclinical models of infantile hypercalcemia type 1.

Strain/Genotype	Sex	25(OH)D_3_ (ng/mL)	24,25(OH)_2_D_3_ (ng/mL)	Ratio 25:24,25	1,24,25(OH)_3_D_3_ (pg/mL)	25(OH)D_3_-26,23-lactone (ng/mL)	1,25(OH)_2_D_3_ (pg/mL)
** *Cyp24a1* ^+/−^ **	m	–	–	–	–	–	-
	f	17.21 ± 2.36	6.68 ± 1.05	2.60 ± 0.10	72.80 ± 4.20	3.84 ± 0.72	27.80 ± 4.20
** *Cyp24a1* ^−/−^ **	m	–	–	–	–	–	–
	f	101.29 ± 20.61 ***	1.45 ± 0.11 ***	70.60 ± 15.7 ^#^	n.d.	0.06 ± 0.02 ^#^	34.00 ± 6.20
**C57bl6-R396^+/+^**	m	19.59 ± 2.90	6.71 ± 0.76	2.91 ± 0.33	57.77 ± 10.88	2.76 ± 0.32	27.39 ± 3.35
	f	18.59 ± 2.50	7.38 ± 0.96	2.53 ± 0.25	66.53 ± 13.18	3.16 ± 0.89	21.28 ± 5.96
**C57bl6-R396^w/w^**	m	110.73 ± 15.40 ^#^	2.59 ± 0.44 ^#^	43.10 ± 4.08 ^#^	17.81 ± 3.53 ^#^	0.07 ± 0.03 ^#^	37.06 ± 8.85
	f	106.16 ± 26.73 ^#^	2.12 ± 0.44 ^#^	49.71 ± 4.37 ^#^	11.09 ± 5.35 ^#^	0.05 ± 0.02 ^#^	37.17 ± 12.12 **
**CD1-R396^+/+^**	m	17.23 ± 3.77	6.31 ± 1.89	2.94 ± 1.07	74.66 ± 34.78	2.24 ± 0.70	45.83 ± 22.74
	f	18.24 ± 4.14	8.80 ± 2.14	2.08 ± 0.19	55.19 ± 17.50	2.56 ± 0.67	27.43 ± 9.77
**CD1-R396^w/w^**	m	81.90 ± 19.39 ^#^	2.26 ± 0.53 ^#^	36.58 ± 5.35 ^#^	15.18 ± 5.84 ^#^	0.06 ± 0.05 ^#^	45.12 ± 16.57
	f	99.16 ± 13.59 ^#^	2.23 ± 0.39 ^#^	44.90 ± 5.53 ^#^	7.49 ± 1.95 ^#^	0.03 ± 0.01 ^#^	29.24 ± 7.98

Results are means ± S.D. Ratio 25:24,25, ratio of 25(OH)_2_D_3_ concentrations over 24,25(OH)_2_D_3_ concentrations. n.d., non-detectable. **, *p* < 0.01; ***, *p* < 0.001; ^#^, *p* < 0.0001 vs. corresponding wild-type sex within strain by 2-way ANOVA and Sidak’s post hoc test.

## Data Availability

Not applicable.

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
