# Peer review of "Vitamin D and Diseases of Mineral Homeostasis: A Cyp24a1 R396W Humanized Preclinical Model of Infantile Hypercalcemia Type 1"

_nutrients, 2022, doi:10.3390/nu14153221_

Round 1

Reviewer 1 Report

The authors report on a mouse model of Cyp24A1 mutation known to cause infantile hypercalcemia in humans

The model largely confirms human data but remarkably

-        The gene expression of cyp24a1 is very high

-        There is remaining  24.25D in serum

-        The severity of the disease is different when expressed in mice with different general background

The work is well described and interesting as can be expected from these groups

Based on human data: are heterozygous mice affected  or not?

Are there signs of extra renal calcifications?

Author Response

NOTE: cited line numbers refer to the version of the manuscript where all changes have been accepted (without tracking)

Reviewer 1

We thank the reviewer for his insightful comments that have allowed us to submit an improved revised manuscript.

Specific points:

1. Based on human data: are the heterozygous mice affected or not?

Reply: the reduced strength of the bone fracture repairing callus was the only phenotypic manifestation that we observed in heterozygous carriers of the mutation. This is now mentioned in the Discussion, lines 352-355 of the manuscript.

2. Are there signs of extra renal calcification?

Reply: we did not investigate for signs of extra renal calcification. This is now specified on lines 206-207 of the revised manuscript.

Reviewer 2 Report

Dear Authors,

congratulations for the quality of your paper and the accuracy of your datas. 

The only suggestions I have to propose are:

- Divide the discussion paragraph in Discussion/Conclusions 

- Is it necessary to write paragraph 2.8 in bold?

- Regarding lines 190, 208, 307, why are data not shown?

Best regards

Author Response

NOTE: cited line numbers refer to the version of the manuscript where all changes have been accepted (without tracking)

Reviewer 2:

We thank the reviewer for his positive and supportive review.

Specific points:

1. Divide the discussion paragraph in Discussion/Conclusions

Reply: we have added a section 5, Conclusions on lines 359-363 of the revised manuscript.

2. Is it necessary to write paragraph 2.8 in bold?

Reply: we apologize for this formatting error that escaped our proofreading prior to submission. This has been corrected and paragraph 2.8 is now unbolded.

3. Regarding lines 190, 208, 307, why are data not shown?

Reply: line 190 referred to genotyping data from the commercial supplier of the mice. We do not feel that the results contribute significantly to the study and have simply deleted the ‘not shown’ mention on this line.

Line 208: we have added the data as Supplementary Figure S1.

Line 307: we have added the data as Supplementary Figure S3.